# BALANCING THE FALSE POSITIVE-NEGATIVE TRADE-OFF TO ENHANCE IMAGE SEGMENTATION

## ABSTRACT

Image segmentation is a fundamental task in computer vision with applications spanning diverse domains, particularly in medical imaging. However, the effectiveness of segmentation techniques often varies across datasets and tasks. For instance, methods like SRL and cLDice focus on segmenting thin tubular structures, while models such as IC-Net are tailored for brain tumor segmentation in MRI scans. Despite the availability of such specialized approaches, there remains a need for a unified framework that can generalize well across different segmentation challenges. In this study, we work on the observation that most of the strategies disproportionately emphasize reducing either False Negatives (FN) or False Positives (FP) and fail to achieve an optimal balance between the two. Building on this observation, we propose a novel method, Supervised Mask Modulation (SMM), that enhances segmentation performance by conditioning the ground truth masks during training to keep a balance between both highly important metrics. Our approach is architecture-agnostic and has been validated on a range of benchmark datasets, consistently outperforming state-of-the-art methods, often achieving significantly better results than the baseline.

## 1 INTRODUCTION

In the ever-evolving field of computer vision, image segmentation remains a critical task with wide-ranging applications in domains such as autonomous driving, industrial inspection, and medical imaging. To tackle the diverse challenges posed by different segmentation problems, a multitude of architectural innovations have been proposed over the years. Classical designs like U-Net (Azad et al., 2024) laid the groundwork for encoder-decoder based segmentation, and have since been extended in numerous directions. Several methods have been developed to address domain-specific challenges, particularly in medical image segmentation. Innovations in loss function such as centerlineDice (clDice) (Shit et al., 2021) and Skeleton Recall Loss (SRL) (Kirchhoff et al., 2024), specifically target segmentation of thin curvilinear structures. On similar lines, Kervadec (Kervadec et al., 2019) utilizes a novel loss function focusing on enhanced boundary predictions. Apart from novel loss functions, architectural novelties have also been introduced, catering to specific segmentation tasks. ICNet (Li et al., 2020) integrated a convolutional network with multiple resolution inputs, designed for accurate tumor core and enhancing region segmentation in brain tumor segmentation tasks. DeepMedic (Kamnitsas et al., 2016) introduced a dual-pathway 3D CNN with dense inference for effective segmentation of small lesions in brain MRI. DUNet (Sheng et al., 2024) introduced constant resolution U-blocks and dense feature connections to effectively detect fine-grained cracks with high accuracy and generalization, even in cluttered scenes.

Amidst the wide array of available segmentation methodologies, selecting an approach well-suited to a specific task can be non-trivial. To address this challenge, we propose **Supervised Mask Modulation (SMM)**, a unified architecture-agnostic strategy designed to enhance segmentation performance across diverse tasks, including those involving complex and irregular structures. In medical image segmentation, a false negative corresponds to a missed abnormality, such as a tumor or lesion not being detected, whereas a false positive refers to incorrectly identifying a healthy region as diseased. Minimizing false negatives is especially critical in clinical settings, as they may lead to missed diagnoses or delayed treatments. Our framework is based on the observation that in medical image segmentation tasks, the False Negative Rate (FNR) is often significantly higher than the False Positive Rate (FPR). This observation is supported by both prior experimental data (see Section 2.1)

and empirical evidence from our current results (see Section 5). But working merely to reduce the FNR can take be equally damaging in terms of the toll bore by other generic metrics(Dice Similarity Coefficient (DSC), Jaccard Similarity Index (JSI), clDice) caused by the imbalance of the FP.

The proposed methodology builds upon the integration of two well-established paradigms: controlling FN-FP balance and mask modulation. While each of these strategies has independently demonstrated effectiveness in improving segmentation performance, their isolated application often limits generalizability across diverse tasks (see Section 2.2). To overcome these limitations, we introduce a unified framework that synergistically combines false negative suppression with mask modulation, thereby enabling a more versatile and robust solution adaptable to a broader spectrum of segmentation challenges.

The proposed framework has been rigorously evaluated on a diverse set of publicly available datasets, each selected to represent distinct and challenging segmentation scenarios. These datasets span a broad spectrum of applications, ranging from irregular and complex structures in histopathology images to object delineation in real-world aerial imagery. This diversity underscores the generalizability and robustness of our approach across varied domains and segmentation tasks. With the proposed SMM framework we make the following contributions:

- **Exploitation of the FP**: Our framework exploits the hypothesis that the number of FN is significantly higher than the number of FP. We attempt to improve the model's performance by introducing intended FP, conditioned by model performance, into the ground truth masks for enhanced training, thereby penalizing the model for missing out class pixels in smaller regions or some structures entirely. The results validate that this strategy tends to bring an overall improvement in the model performance.

- **Multi-Class Compatibility**: The framework has proved effective across diverse datasets, demonstrating improved performance on both binary and multi-class segmentation tasks. Its versatility makes it applicable to a wide range of imaging scenarios.

- **Architecture Agnostic**: The framework does not include making alterations to any particular architecture. rather, it proposes in a change in the training paradigm of models and can therefore integrate seamlessly with a wide range of pipelines across different model architectures.

## 2 HIGHLIGHTING THE PROBLEM

### 2.1 THE EXAGGERATED FALSE NEGATIVES

While numerous studies have proposed segmentation techniques tailored to specific tasks, several underlying principles emerge that are broadly applicable across diverse segmentation problems:

1. De Rosa et al. (2024) observed that although their U-Net-based ensemble achieved high precision, it exhibited low recall, indicating that while false positives (FP) were reduced, false negatives (FN) remained predominant.

2. In a teacher-student weakly supervised setup for colon polyp segmentation, Jia et al. (2024) reported that the segmentation outputs "present a quite high FNR inside the polyp area."

3. By modulating the Tversky-loss parameter $\beta$, Do et al. (2020) highlighted the FNR-FPR trade-off, noting that "as $\beta$ increased, the false-positive rate systematically decreased while the false-negative rate systematically increased."

These observations, corroborated by additional studies (Delgado et al., 2024; Luo et al., 2023), demonstrate that FNR often substantially exceeds FPR in many segmentation tasks. Our own evaluation of U-Net models, reported in Table 1, further confirms this trend.

This phenomenon can be intuitively explained in medical imaging tasks such as brain tumor or lesion segmentation, where the foreground region typically occupies only a small fraction of the image relative to the background. Such an imbalance hampers the model's ability to comprehensively capture the foreground. From a theoretical standpoint, standard objectives such as cross-entropy minimize the expected misclassification rate under maximum likelihood estimation, implicitly assuming equal costs for false positives and false negatives. Consequently, they fail to emphasize recall

in settings where false negatives are more critical. Evaluation metrics in many domains, including medical imaging, are based on precision and recall rather than accuracy, and these metrics are not aligned with the likelihood training criterion (Goodfellow et al., 2016, p. 265). This misalignment often leads to models that prioritize precision over recall, exacerbating the FNR. Our method leverages this insight by guiding the model to predict additional positives, thereby reducing FNR while minimally affecting FPR, ultimately achieving an optimal trade-off between these inversely related metrics.

## 2.2 RELATED WORK

Enhancing segmentation performance by mitigating FN, often reflected as improved recall, has been a central focus of recent studies. A common approach involves modifying loss functions to increase model sensitivity, particularly in imbalanced or complex medical datasets. For instance, Xiang et al. (2019a;b) designed loss functions to enhance reliability, while Chan et al. (2020) employed maximum likelihood estimation with Bayesian decision theory to better handle underrepresented classes. Other methods, including Zhong et al. (2021) and Kervadec et al. (2019), introduce pixel-wise or contour-based losses specifically aimed at reducing FN in fine structures. In this context, loss functions such as Tversky Salehi et al. (2017) and Focal Lin et al. (2017) have also been widely adopted: the Tversky loss allows explicit weighting of false negatives relative to false positives, while Focal loss down-weights easy examples to focus training on harder, underrepresented regions, making them particularly effective in medical segmentation tasks. Beyond loss design, architectural innovations such as PatchRefineNet (Nagendra & Kifer, 2024) refine outputs by correcting spatial biases in logits, and depth-based strategies (Maag, 2021; Maag & Rottmann, 2022) further improve recall. However, these architectural solutions often incur additional computational complexity, rendering loss-based approaches a more lightweight and widely adopted alternative.

Mask transformations have also been explored to enhance segmentation. Skeletonization transforms have been utilized for the detection of fine tubular structures, with a focus on preserving topological integrity (Kirchhoff et al., 2024; Shit et al., 2021). Similarly, Kats et al. (2019) introduced a soft-labeled mask combined with soft dice loss for lesion segmentation, while Vasudeva et al. (2024) employed geodesic distance transforms to assign soft labels near boundaries.

Building on these efforts, we propose a novel mask transformation strategy guided by model-predicted false negatives, complemented by tailored training mechanisms. This approach broadens the applicability of our methodology while directly addressing the FNR-FPR trade-off in segmentation tasks.

## 3 METHODOLOGY

Given that FPR values are consistently lower than FNR in medical imaging tasks, we leverage this asymmetry to guide our approach. Specifically, the framework introduces controlled FP regions in the ground truth masks to encourage the model to predict positives in previously missed areas. Implementation details are discussed in subsequent sections.

### 3.1 MISS-AWARE MASK MODULATION (MAMM)

---
**Algorithm 1** Miss-Aware Mask Modulation (MAMM)

---
**Require:** Prediction $\hat{\mathbf{Y}}$, Ground truth $\mathbf{Y}$
1: $\mathbf{FN} \leftarrow (\mathbf{Y} - \hat{\mathbf{Y}}) > 0$         ▷ Extract false negatives
2: $\mathbf{U} \leftarrow \text{Dilate}(\mathbf{FN})$         ▷ Expand missed regions
3: $\mathbf{Y^M} \leftarrow \mathbf{U} \cup \mathbf{Y}$        ▷ Generate modulated mask
4: **return** $\mathbf{Y^M}$

---

FN are defined as regions in the ground truth mask that belong to the foreground but were incorrectly predicted as background by the segmentation model. To extract these missed regions, we compute the difference between the ground truth mask, $\mathbf{Y}$, and the predicted mask, $\hat{\mathbf{Y}}$:

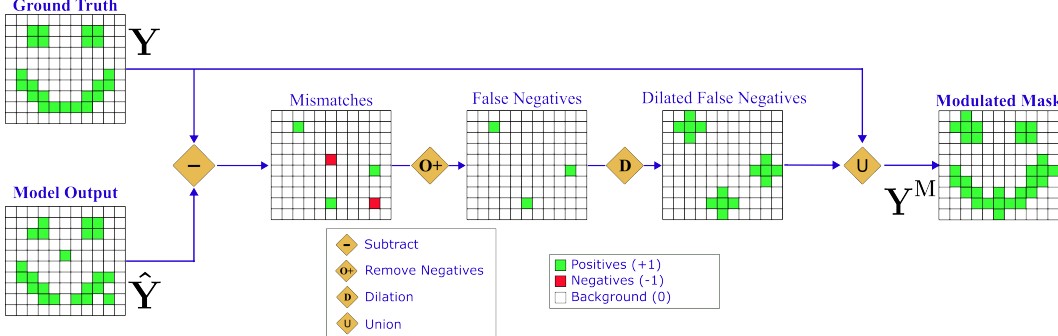

Figure 1: **Miss-Aware Mask Modulation** The mask modulation process begins by subtracting the predicted mask from the ground truth to identify misclassified pixels. Only the false negatives are retained, dilated, and combined with the original ground truth to generate the updated modulated mask.

$$\mathbf{FN} = (\mathbf{Y} - \hat{\mathbf{Y}}) > 0,$$

where positive values correspond to false negatives, negative values correspond to false positives, and correctly classified pixels (true positives and true negatives) are reduced to zero. We then retain only the positive entries corresponding to the FN.

**Dilation:** To emphasize regions overlooked by the model, the FN mask is dilated with a diamond-shaped kernel of radius 2, and its union with the ground truth yields the updated modulated mask. This operation is performed independently for each class to construct the final mask.

This modulation strategy, termed **Miss-Aware Mask Modulation (MAMM)**, adaptively updates the mask to reflect the model's current errors while remaining anchored to the original ground truth. By refreshing the modulated masks at each epoch, the procedure ensures that training consistently targets the most recent false negatives (Figure 1).

### 3.2 TRAINING ALGORITHM

Algorithm 1 provides a mechanism to enhance focus on regions prone to being missed. We propose two strategies to leverage this transformation during training, differing in the degree of penalization applied to the model. These can be interpreted as 'hard' and 'soft' approaches to mask modulation, which are described in detail in the subsequent subsections.

---

**Algorithm 2** Supervised Mask Modulation *v1*

---

**Require:** Input $\mathbf{X}$, Ground Truth $\mathbf{Y}$, and Modulated Mask $\mathbf{Y_0^M} = \mathbf{Y}$
1: **for** each epoch ($t$) **do**
2:     $\hat{\mathbf{Y}}_\mathbf{t} \leftarrow \text{Model}(\mathbf{X})$
3:     $\mathscr{L} \leftarrow \mathscr{L}_{vanilla}(\hat{\mathbf{Y}}_\mathbf{t}, \mathbf{Y}) + \mathscr{L}_{ESL}(\hat{\mathbf{Y}}_\mathbf{t}, \mathbf{Y_t^M})$
4:     Backpropagate loss $\mathscr{L}$
5:     $\tilde{\mathbf{Y}}_{\mathbf{t+1}} \leftarrow \text{MAMM}(\hat{\mathbf{Y}}_\mathbf{t}, \mathbf{Y})$
6: **end for**

---

#### 3.2.1 SUPERVISED MASK MODULATION *v1*

In this variant, we propose a specialized loss function, Elevated Senstivity Loss (ESL), explicitly designed to impose a strong penalty on FN in the model's predictions. Its primary objective is to ensure that small or subtle regions are accurately detected and not overlooked. This is achieved by incorporating the count of FN directly into the denominator of the loss formulation. To maintain scale invariance and normalize the contribution of each pixel, the total number of pixels, being a

constant, is also included in the denominator. The explicit focus on penalizing missed detections characterizes this approach as a hard penalization strategy, motivating its designation as the hard training algorithm for SMM.

Let $Y$ denote the ground truth mask and $\hat{Y}$ denote the predicted mask, each consisting of $N$ pixels indexed by $i \in \Omega$, where $\Omega$ is the set of all pixel locations. Let $y_i \in \{0, 1\}$ and $\hat{y}_i \in [0, 1]$ denote the values of the $i^{\text{th}}$ pixel in $Y$ and $\hat{Y}$, respectively. We define the **Elevated Sensitivity Loss (ESL)** as:

$$\mathscr{L}_{\text{ESL}} = -\frac{\sum_{i\in\Omega} y_i\,\hat{y}_i}{N + \sum_{i\in\Omega} y_i(1-\hat{y}_i)} \tag{1}$$

where:

- $N = |\Omega|$ is the total number of pixels in the mask.
- $y_i \in \{0, 1\}$ is the ground truth value of the $i^{\text{th}}$ pixel.
- $\hat{y}_i \in [0, 1]$ is the predicted value of the $i^{\text{th}}$ pixel.

The multiplicative factor of $N$ acts as a normalization term for the loss, ensuring scale consistency. Its significance is discussed in further detail in Appendix C.

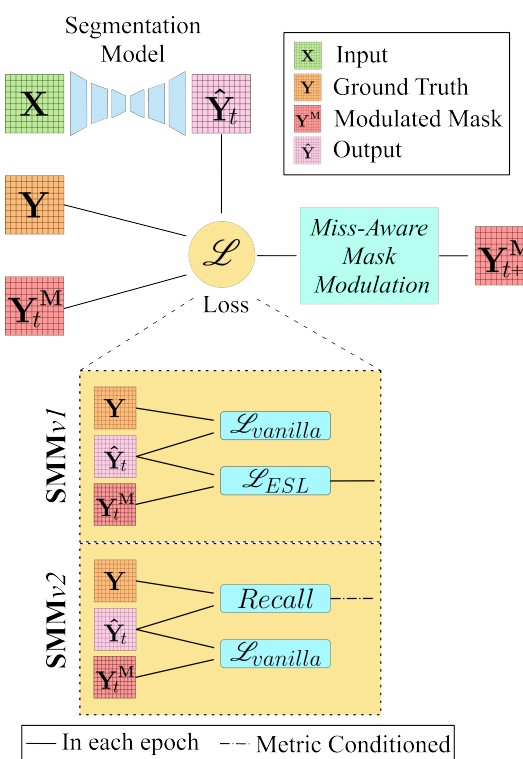

Figure 2: **Supervised Mask Modulation.** Given an input $\mathbf{X}$, the segmentation model predicts $\hat{\mathbf{Y}}$, which is compared with the ground truth $\mathbf{Y}$ and modulated mask $\mathbf{Y_t^M}$ to compute the loss. This loss updates the modulated mask for subsequent epochs, yielding $\mathbf{Y_{t+1}^M}$. Both SMM variants employ MAMM to generate these masks.

The ESL loss is applied in conjunction with modulated masks generated from Algorithm 1. Training begins with a warm-up phase, defaulted to 20% of total epochs, during which neither mask updates nor ESL computation occurs, allowing the model to learn global structures. Post pretraining, masks are updated each epoch, and ESL is computed using the modulated masks, while standard loss functions operate on the original ground truth. The final loss is the sum of both, ensuring the model follows the general learning trajectory while explicitly penalizing FN.

### 3.2.2 SUPERVISED MASK MODULATION v2

We propose an adaptive training strategy that modulates the mask based on the model's performance, measured via the *recall metric* after each epoch. Let the model first undergo pretraining for 20% of the total epochs. We consistently store the recall values in a fixed-length queue of size $L$.

To assess the trend of model performance, we compute the gradient of the best-fit line for recall over epochs, referred to as $\beta$:

$$\beta = \frac{\text{Cov}(\mathbf{x}, \mathbf{y})}{\text{Var}(\mathbf{y})}, \tag{2}$$

with

$$\mathbf{x} = [x_1, x_2, \ldots, x_L]^\top, \quad x_i : \text{recall at epoch } i,$$
$$\mathbf{y} = [1, 2, \ldots, L]^\top, \qquad y_i : \text{epoch index } i,$$

where $\text{Var}(\mathbf{y})$ is the variance of $\mathbf{y}$ and $\text{Cov}(\mathbf{x}, \mathbf{y})$ the covariance between $\mathbf{x}$ and $\mathbf{y}$.

A small or negative $\beta$ indicates stagnation or decline in performance, prompting updates to the modulated mask. When $\beta$ falls below a threshold $\gamma$, controlled FP are introduced near the true boundary via dilation (Algorithm 3), while FN are consistently computed against the original ground truth. To prevent excessive mask expansion, prior modulations are cleared before new ones are applied. This thresholding mechanism balances exploration and consolidation by adapting to gradient magnitude, drawing inspiration from adaptive thresholding in semi-supervised learning (Xu et al., 2021): gradients above $\gamma$ suppress modulation, reinforcing confident regions, whereas those below $\gamma$ trigger updates to redirect learning toward uncertain or overlooked areas. Recall is employed solely for evaluation and excluded from optimization, and $\gamma$ is linearly decayed during training to reflect the model's evolving identification of novel positives. Unlike earlier variants, this approach dispenses with explicit penalization, motivating its designation as the soft training algorithm for SMM (Algorithm 4).

---

**Algorithm 3** UpdateMask

---

**Require:** Queue $Q$, Prediction $\hat{\mathbf{Y}}_{\mathbf{t}}$, Ground truth $\mathbf{Y}$, and Modulated Mask $\mathbf{Y}_{\mathbf{t}}^{\mathbf{M}}$
 1: **Retrieve** threshold parameter $\gamma$
 2: Compute recall $r$ between $\mathbf{Y}$ and $\hat{\mathbf{Y}}_{\mathbf{t}}$:
$$r = \frac{\sum(\mathbf{Y} \wedge \hat{\mathbf{Y}}_{\mathbf{t}})}{\sum \mathbf{Y}}$$
 3: Append $r$ to queue $Q$
 4: Compute gradient $\beta$ from queue $Q$ using Eq. 2
 5: **if** $\beta < \gamma$ **then**
 6: $\quad \mathbf{Y}_{\mathbf{t+1}}^{\mathbf{M}} \leftarrow \text{MAMM}(\hat{\mathbf{Y}}_{\mathbf{t}}, \mathbf{Y})$
 7: **else**
 8: $\quad \mathbf{Y}_{\mathbf{t+1}}^{\mathbf{M}} \leftarrow \mathbf{Y}_{\mathbf{t}}^{\mathbf{M}}$
 9: **end if**
10: **return** $\mathbf{Y}_{\mathbf{t+1}}^{\mathbf{M}}$

---

---

**Algorithm 4** Supervised Mask Modulation *v2*

---

**Require:** Input $\mathbf{X}$, Ground Truth $\mathbf{Y}$, and Modulated Mask $\mathbf{Y}_{\mathbf{0}}^{\mathbf{M}} = \mathbf{Y}$
 1: **Initialize:** Empty queue $Q$ with fixed length $L$
 2: **for** each epoch $t$ **do**
 3: $\quad \hat{\mathbf{Y}}_{\mathbf{t}} \leftarrow \text{Model}(\mathbf{X})$
 4: $\quad \mathscr{L} \leftarrow \text{Loss}(\hat{\mathbf{Y}}_{\mathbf{t}}, \mathbf{Y}_{\mathbf{t}}^{\mathbf{M}})$
 5: $\quad$ Backpropagate using $\mathscr{L}$
 6: $\quad \mathbf{Y}_{\mathbf{t+1}}^{\mathbf{M}} \leftarrow \text{UPDATEMASK}(Q, \hat{\mathbf{Y}}_{\mathbf{t}}, \mathbf{Y}, \mathbf{Y}_{\mathbf{t}}^{\mathbf{M}})$
 7: **end for**

---

## 4 EXPERIMENTAL SETUP

### 4.1 MODEL CONFIGURATIONS

We evaluated model performance using a standard U-Net (Ronneberger et al., 2015) pipeline with four encoder-decoder stages. Each stage consists of two convolutional layers with batch normalization and ReLU activation. Decoder features are upsampled via transposed convolutions and fused with encoder outputs through skip connections, preserving spatial detail.

Models were trained with an initial learning rate of 0.1 and linear decay, using a batch size of 4. Epochs were dataset-specific to ensure convergence. All experiments were run on NVIDIA Tesla T4 and GeForce GTX 1080 Ti GPUs.

### 4.2 TRAINING SETUP

We evaluate our models using U-Net as the base architecture, comparing against strong, architecture-agnostic baselines. All experiments were repeated over five random seeds for significance analysis.

Table 1: **Test set metrics** of U-Net models trained using different strategies. SMM*v1* and SMM*v2* present results for both versions of our proposed framework. **Bold** shows the best metric value, while Underline shows the second best metric.

| Method | DSC $\uparrow$ | clDice $\uparrow$ | JSI $\uparrow$ | FNR $\downarrow$ | FPR $\downarrow$ |
|---|---|---|---|---|---|
| **BoMBR** (Raina et al., 2024) | | | | | |
| Vanilla U-Net | $66.02_{\pm 2.11}$ | $63.57_{\pm 1.76}$ | $56.42_{\pm 2.44}$ | $26.04_{\pm 1.82}$ | $7.58_{\pm 0.81}$ |
| U-Net + SRL | $66.84_{\pm 2.21}$ | $63.54_{\pm 1.36}$ | $57.24_{\pm 2.41}$ | $25.78_{\pm 2.07}$ | $7.29_{\pm 0.95}$ |
| U-Net + BL | $\underline{67.09}_{\pm 1.06}$ | $64.15_{\pm 1.29}$ | $\underline{57.80}_{\pm 1.04}$ | $26.11_{\pm 1.15}$ | $\underline{7.23}_{\pm 0.38}$ |
| U-Net + Tversky | $66.64_{\pm 0.96}$ | $63.76_{\pm 0.39}$ | $57.18_{\pm 1.02}$ | $\underline{25.87}_{\pm 1.25}$ | $7.37_{\pm 0.25}$ |
| U-Net + Focal | $65.96_{\pm 1.14}$ | $\mathbf{65.53}_{\pm \mathbf{0.64}}$ | $56.37_{\pm 1.17}$ | $26.05_{\pm 1.32}$ | $7.49_{\pm 0.64}$ |
| SMM*v1* | $66.82_{\pm 1.16}$ | $64.12_{\pm 0.95}$ | $57.37_{\pm 1.38}$ | $25.92_{\pm 0.77}$ | $7.27_{\pm 0.39}$ |
| SMM*v2* | $\mathbf{67.46}_{\pm \mathbf{1.24}}$ | $\underline{64.42}_{\pm 0.64}$ | $\mathbf{57.96}_{\pm \mathbf{1.13}}$ | $\mathbf{24.73}_{\pm \mathbf{1.14}}$ | $\mathbf{7.09}_{\pm \mathbf{0.39}}$ |
| **DRIVE** (Hassan et al., 2015) | | | | | |
| Vanilla U-Net | $79.63_{\pm 1.45}$ | $83.48_{\pm 1.84}$ | $66.21_{\pm 1.98}$ | $21.51_{\pm 2.00}$ | $2.55_{\pm 0.10}$ |
| U-Net + SRL | $80.01_{\pm 0.47}$ | $\underline{84.27}_{\pm 0.81}$ | $66.72_{\pm 0.65}$ | $\mathbf{18.85}_{\pm \mathbf{0.97}}$ | $2.97_{\pm 0.08}$ |
| U-Net + BL | $79.72_{\pm 1.74}$ | $83.36_{\pm 1.48}$ | $66.35_{\pm 2.39}$ | $23.40_{\pm 1.13}$ | $\mathbf{2.12}_{\pm \mathbf{0.39}}$ |
| U-Net + Tversky | $79.79_{\pm 0.34}$ | $83.70_{\pm 0.75}$ | $66.41_{\pm 0.48}$ | $21.26_{\pm 0.52}$ | $2.55_{\pm 0.11}$ |
| U-Net + Focal | $\underline{80.22}_{\pm 0.87}$ | $84.24_{\pm 0.88}$ | $\underline{67.02}_{\pm 1.22}$ | $\underline{20.84}_{\pm 1.03}$ | $2.48_{\pm 0.09}$ |
| SMM*v1* | $\mathbf{80.64}_{\pm \mathbf{1.30}}$ | $\mathbf{84.42}_{\pm \mathbf{1.51}}$ | $\mathbf{67.62}_{\pm \mathbf{1.83}}$ | $20.98_{\pm 2.08}$ | $\underline{2.31}_{\pm 0.26}$ |
| SMM*v2* | $78.93_{\pm 0.68}$ | $82.71_{\pm 0.94}$ | $65.24_{\pm 0.92}$ | $21.53_{\pm 1.08}$ | $2.79_{\pm 0.09}$ |
| **Cracks** (Tomaszkiewicz & Owerko, 2023) | | | | | |
| Vanilla U-Net | $\underline{64.57}_{\pm 0.87}$ | $\underline{74.92}_{\pm 1.15}$ | $\underline{51.20}_{\pm 0.80}$ | $31.39_{\pm 1.22}$ | $\underline{0.33}_{\pm 0.01}$ |
| U-Net + SRL | $62.51_{\pm 3.31}$ | $71.93_{\pm 5.15}$ | $49.12_{\pm 3.36}$ | $\mathbf{29.69}_{\pm \mathbf{0.80}}$ | $0.44_{\pm 0.13}$ |
| U-Net + BL | $64.05_{\pm 0.93}$ | $74.73_{\pm 0.82}$ | $50.82_{\pm 0.93}$ | $33.15_{\pm 0.66}$ | $\mathbf{0.31}_{\pm \mathbf{0.01}}$ |
| U-Net + Tversky | $64.33_{\pm 0.36}$ | $74.75_{\pm 0.35}$ | $51.02_{\pm 0.24}$ | $\underline{30.97}_{\pm 0.50}$ | $0.34_{\pm 0.00}$ |
| U-Net + Focal | $62.10_{\pm 3.13}$ | $71.15_{\pm 4.93}$ | $48.91_{\pm 3.25}$ | $33.18_{\pm 3.97}$ | $0.40_{\pm 0.19}$ |
| SMM*v1* | $\mathbf{64.74}_{\pm \mathbf{0.20}}$ | $\mathbf{75.35}_{\pm \mathbf{0.34}}$ | $\mathbf{51.44}_{\pm \mathbf{0.21}}$ | $31.16_{\pm 0.58}$ | $\underline{0.33}_{\pm 0.01}$ |
| SMM*v2* | $62.93_{\pm 2.73}$ | $72.64_{\pm 4.50}$ | $49.56_{\pm 2.80}$ | $33.08_{\pm 2.82}$ | $0.33_{\pm 0.02}$ |
| **Drone**[1] | | | | | |
| Vanilla U-Net | $49.58_{\pm 2.54}$ | $44.44_{\pm 2.34}$ | $39.95_{\pm 2.04}$ | $29.47_{\pm 2.76}$ | $6.32_{\pm 0.25}$ |
| U-Net + SRL | $48.92_{\pm 1.45}$ | $43.92_{\pm 1.38}$ | $39.16_{\pm 1.02}$ | $29.69_{\pm 1.23}$ | $6.40_{\pm 0.24}$ |
| U-Net + BL | $45.37_{\pm 8.06}$ | $40.13_{\pm 7.81}$ | $37.76_{\pm 6.60}$ | $36.23_{\pm 7.17}$ | $6.73_{\pm 1.28}$ |
| U-Net + Tversky | $49.64_{\pm 2.37}$ | $44.49_{\pm 2.23}$ | $40.23_{\pm 1.85}$ | $30.54_{\pm 3.34}$ | $6.13_{\pm 0.29}$ |
| U-Net + Focal | $47.19_{\pm 2.99}$ | $42.47_{\pm 3.18}$ | $38.08_{\pm 2.37}$ | $32.59_{\pm 3.33}$ | $6.44_{\pm 0.25}$ |
| SMM*v1* | $\underline{50.49}_{\pm 1.72}$ | $\underline{45.45}_{\pm 1.91}$ | $\underline{40.89}_{\pm 1.46}$ | $\underline{29.20}_{\pm 1.79}$ | $\underline{6.07}_{\pm 0.28}$ |
| SMM*v2* | $\mathbf{51.34}_{\pm \mathbf{2.39}}$ | $\mathbf{46.21}_{\pm \mathbf{2.09}}$ | $\mathbf{41.61}_{\pm \mathbf{2.31}}$ | $\mathbf{27.70}_{\pm \mathbf{2.20}}$ | $\mathbf{5.93}_{\pm \mathbf{0.33}}$ |

The baselines are: **Vanilla U-Net** (trained with Dice loss (Dice)+Categorical Cross Entropy loss (CCE)), **U-Net+SRL** (Vanilla U-Net with SRL), and **U-Net+BL** (Vanilla U-Net with Boundary Loss (BL)).We also integrate the **Tversky Loss** Salehi et al. (2017) and the **Focal Loss** Lin et al. (2017), both of which introduce mechanisms to control the relative weighting of false negatives, thereby enhancing performance on underrepresented or difficult-to-detect structures.

Since CCE fails under overlapping class regions induced by MAMM, SMM*v2* replaces it with class-wise Binary Cross-Entropy to support multi-label pixels. We set the queue length $L = 15$ and $\gamma$ to the mean of $\beta$ values from pretraining epochs. For baseline evaluation, using Tversky Loss, we set the loss parameter $\alpha = 0.3$.

Evaluation employed complementary metrics: (i) **Overlap** (DSC, JSI), (ii) **Topology** (clDice), and (iii) **Error** (FNR, FPR), covering accuracy, structural preservation, and under/over-segmentation tendencies.

# 5 RESULTS AND DISCUSSION

## 5.1 EVALUATION PROTOCOL

To assess the robustness and generalizability of our approach, we validated the method across a diverse set of benchmark datasets. Following the recommendations of *The Machine Learning Reproducibility Checklist* (Pineau et al., 2021), we attempted to mitigate stochastic effects in training and ensure reproducibility by repeating each experimental configuration using five fixed random seeds. Reported results are expressed as mean $\pm$ standard deviation across these runs, providing a reliable estimate of model performance while attributing observed differences to methodological improvements rather than random variations in initialization or data shuffling.

## 5.2 METRIC-LEVEL INSIGHTS

Table 1 shows that SRL consistently reduces FNR, often with a moderate increase in FPR relative to Vanilla U-Net, while BL shows the opposite trend. For example, on the BoMBR dataset, SRL lowers FNR from 26.04% to 25.78%, whereas BL slightly increases it to 26.11%. On DRIVE, SRL achieves the lowest FNR of 18.85%, compared to 21.51% for Vanilla U-Net, while BL attains the lowest FPR of 2.12%.

Loss-based strategies, such as Tversky and Focal, also aim to improve segmentation under class imbalance. Tversky loss explicitly weights false negatives relative to false positives, achieving moderate FNR reductions (like 25.87% on BoMBR, 21.26% on DRIVE), while Focal loss emphasizes hard-to-classify pixels, improving metrics like clDice (65.53% on BoMBR, 84.24% on DRIVE).

However, our SMM variants consistently outperform these specialized losses across both error rates and structural metrics, attaining the best Dice scores in all the datasets - BoMBR (67.46%), Drive (80.64%), Cracks (64.74%), and Drone (51.34%). In datasets with significant structural complexity (BoMBR and Drone), SMM*v2* achieves the lowest FNR - BoMBR (24.73%) and Drone (27.70%), while simultaneously reducing FPR to 7.09% and 5.93%, respectively. For the remaining datasets, the baseline methods struggle with the delicate trade-off between FPR and FNR, tending to skew toward one side. In contrast, SMM*v1* maintains a more stable balance, resulting in improved structural metrics- Drive(84.42% clDice, 67.62% JSI) and Cracks(75.35% clDice, 51.44% JSI). Even in cases where it is not the absolute best, SMM regularly ranks second, highlighting reliable performance gains.

Importantly, our aim is not to achieve the lowest FNR at the cost of increased FPR, or vice versa, but rather to efficiently balance the two to achieve overall optimal segmentation results. In this context, SMM fulfills its design objective, delivering balance in both error rates while improving structural and overlap metrics across datasets.

Statistical significance was evaluated as described in Appendix B, for pairwise comparisons with the strongest baselines. Distinct variant-specific trends across dataset categories are further analyzed in Appendix E.

## 5.3 ARCHITECTURE-AGNOSTIC DEPLOYMENT

Our training framework is designed to be independent of architecture-specific features, enabling robust deployment across diverse segmentation networks. Both variants of SMM employ a unified mask modulation strategy with a generalizable training procedure that can be applied to any segmentation architecture. The effectiveness of this approach is further demonstrated in Appendix D, where we report results on SegNet (Badrinarayanan et al., 2017), showing that SMM maintains strong performance even on architectures not seen during primary experiments.

---

[1]http://dronedataset.icg.tugraz.at/

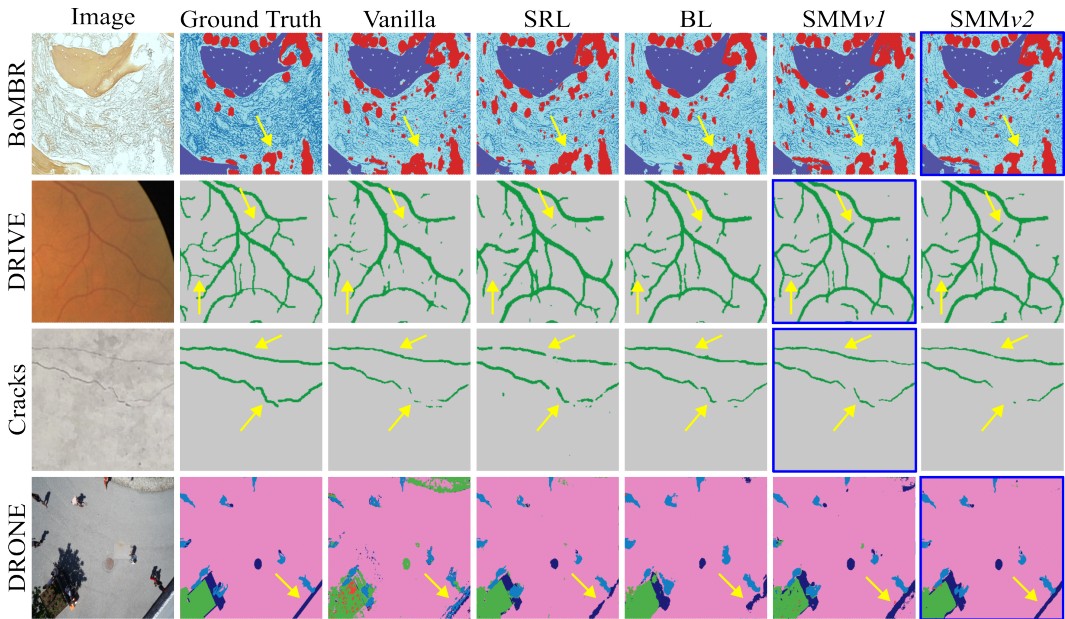

Figure 3: **Visual Results** The figure presents a sample test output of each model for all the utilized datasets. A blue bounding box marks the better-performing version of SMM. The yellow arrows highlight the region where the outputs of different models differ for the given sample.

# 6 CONCLUSION

Despite the proliferation of task-specific segmentation techniques, there remains a pressing need for a unified training paradigm that delivers consistent performance across heterogeneous tasks and domains. In this work, we introduced Supervised Mask Modulation (SMM), an architecture-agnostic segmentation training strategy designed to optimize the balance between FN and FP, thereby enhancing segmentation fidelity.

At the core of SMM is the novel mask transformation, Miss-Aware Mask Modulation (MAMM), derived from model-predicted FN regions, which is combined with two complementary training strategies to reinforce model learning. This approach consolidates the otherwise fragmented landscape of segmentation methods into a generalizable framework. Extensive validation on multiple publicly available datasets, alongside comparisons with state-of-the-art baselines and evaluations across different network architectures, demonstrates that SMM consistently achieves superior performance in both overlap and structural metrics.

Overall, SMM provides a robust and practical methodology for real-world segmentation tasks, achieving an effective trade-off between FNR and FPR, and highlighting the potential of unified, task-agnostic training strategies in advancing segmentation performance across diverse applications.

# 7 REPRODUCIBILITY STATEMENT

We have taken several steps to ensure that our work is fully reproducible. The main paper provides complete descriptions of the model architecture, training objectives, and evaluation setup in Section 4. It contains the information on all hyperparameters and implementation details. We also include a detailed explanation of all the datasets utilized for this study in the Appendix A. We provide the reference used for the implementation of SegNet in Appendix D. Finally, we provide an anonymous link to the source code, along with scripts to run training and evaluation: **Anonymized GitHub**. We will add the actual GitHub link in the Camera-Ready version of the manuscript.

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
