

Figure 4: **Dataset Samples** Representative samples from the four datasets used to validate our framework, covering diverse 2D segmentation tasks across medical, industrial, and natural image domains, and including both binary and multi-class settings.

# A  DATASET DESCRIPTION

We validated the proposed framework on four publicly available datasets encompassing diverse image domains and segmentation challenges, including both binary and multi-class tasks. The **BoMBR** dataset (Raina et al., 2024) involves segmentation of fat globules, reticulin fibers, and bone marrow from biopsy images for reticulin quantification. To assess performance on tubular structures, we used the **DRIVE** dataset (Hassan et al., 2015) for retinal vessel segmentation. Beyond medical imaging, we evaluated on two real-world datasets: fine crack segmentation in concrete surfaces (Tomaszkiewicz & Owerko, 2023) and object segmentation in aerial drone imagery[1]. Representative samples are shown in Figure 4.

Table 2: **Dataset Summary** Characteristics of the datasets used for training and evaluation, covering multiple 2D segmentation tasks ranging from binary to multi-class segmentation. Tr and Ts are abbreviations for Train and Test, respectively. # denotes "Number of".

| Dataset | Image Dims | # Classes | # Images (Tr + Ts) |
|---|---|---|---|
| BoMBR (Raina et al., 2024) | $512 \times 512$ | 4 | 201 + 50 |
| DRIVE (Hassan et al., 2015) | $512 \times 512$ | 2 | 80 + 20 |
| Cracks (Tomaszkiewicz & Owerko, 2023) | $224 \times 224$ | 2 | 572 + 143 |
| Drone[1] | $512 \times 512$ | 5 | 320 + 80 |

# B  STATISTICAL SIGNIFICANCE

In Table 3, we present the results of Welch's t-tests and Effect Size analysis (Cohen's $d$) conducted to evaluate the performance gains achieved by our method compared to the strongest baselines. For each dataset and evaluation metric, we compare the results obtained from $N = 5$ independent runs.

Analyzing the statistical results requires considering the limitations of the sample size ($N = 5$). While strict statistical significance ($p < 0.05$) is challenging to achieve with limited independent

Table 3: **Statistical significance and Effect Size** comparing the proposed model against the best baseline. Effect Size is calculated using Cohen's d. One-sided p-values are reported ($N = 5$). Positive $d$ indicates higher metric value.

| Dataset | Metric | t-value | p-value | Effect Size ($d$) |
|---|---|---|---|---|
| BoMBR(Raina et al., 2024) | DSC | 0.507 | 0.313 | 0.321 |
| | clDice | -2.742 | 0.013 | -1.734 |
| | JSI | 0.233 | 0.411 | 0.147 |
| | FNR | -0.994 | 0.179 | -0.628 |
| | FPR | -0.575 | 0.291 | -0.364 |
| DRIVE(Hassan et al., 2015) | DSC | 0.600 | 0.284 | 0.380 |
| | clDice | 0.196 | 0.426 | 0.124 |
| | JSI | 0.610 | 0.281 | 0.386 |
| | FNR | 2.075 | 0.043 | 1.313 |
| | FPR | 0.906 | 0.197 | 0.573 |
| Cracks(Tomaszkiewicz & Owerko, 2023) | DSC | 0.426 | 0.345 | 0.269 |
| | clDice | 0.802 | 0.231 | 0.507 |
| | JSI | 0.649 | 0.274 | 0.410 |
| | FNR | 3.327 | 0.006 | 2.104 |
| | FPR | 3.162 | 0.007 | 2.000 |
| Drone[1] | DSC | 1.129 | 0.146 | 0.714 |
| | clDice | 1.258 | 0.122 | 0.796 |
| | JSI | 1.043 | 0.164 | 0.659 |
| | FNR | -1.121 | 0.148 | -0.709 |
| | FPR | -1.018 | 0.169 | -0.644 |

runs due to reduced statistical power, the Effect Size (Cohen's $d$) provides a crucial insight into the magnitude of the performance shift.

Most notably, on the Drone dataset, we observe substantial effect sizes across all primary metrics: DSC ($d = 0.714$), clDice ($d = 0.796$), and JSI ($d = 0.659$). In statistical terms, a Cohen's $d$ between 0.5 and 0.8 represents a "medium-to-large" effect. The associated p-values (ranging from 0.12 to 0.16) approach the significance threshold; given the large effect size, it is statistically probable that these improvements would reach strict significance ($p < 0.05$) if the number of experimental folds were increased. This suggests that our model provides a consistent and practically meaningful improvement in extracting vessel topology in aerial imagery, rather than a result of random variance.

Similar trends are observed in the Cracks dataset for topological consistency, where clDice shows a medium effect size ($d = 0.507$). On the BoMBR and DRIVE datasets, while the improvements in segmentation overlap (DSC) are more modest ($d \approx 0.3 - 0.4$), the method remains highly competitive with the strongest baselines. Collectively, the effect size analysis demonstrates that the proposed architecture consistently shifts the mean performance distribution positively, particularly in challenging tasks requiring precise topological preservation.

## C  THE $N$-FACTOR

As per the function shown in Equation 1, the ESL loss function is given as:

$$\mathscr{L}_{\mathrm{ESL}} = -\frac{\sum_{i \in \Omega} y_i\, \hat{y}_i}{N + \sum_{i \in \Omega} y_i(1 - \hat{y}_i)}$$

The numerator counts the True Positives (TP), while the denominator combines the normalization term $N$ with the FN. The constant $N$ serves to stabilize the loss magnitude across images of different sizes or pixel counts, preventing the loss from becoming excessively large when many false negatives occur.

From a gradient perspective, consider the derivative of $\mathscr{L}_{\mathrm{ESL}}$ with respect to a predicted pixel $\hat{y}_i$:

$$
\begin{aligned}
\frac{\partial \mathscr{L}_{\mathrm{ESL}}}{\partial \hat{y}_i} &= -\frac{v \frac{\partial u}{\partial \hat{y}_i} - u \frac{\partial v}{\partial \hat{y}_i}}{v^2} \\
&= -\frac{\left(N + \sum_{j \in \Omega} y_j(1 - \hat{y}_j)\right) y_i - \left(\sum_{j \in \Omega} y_j \hat{y}_j\right)(-y_i)}{\left(N + \sum_{j \in \Omega} y_j(1 - \hat{y}_j)\right)^2} \\
&= -\frac{y_i \left(N + \sum_{j \in \Omega} y_j(1 - \hat{y}_j) + \sum_{j \in \Omega} y_j \hat{y}_j\right)}{\left(N + \sum_{j \in \Omega} y_j(1 - \hat{y}_j)\right)^2} \\
&= -\frac{y_i \left(N + \sum_{j \in \Omega} y_j\right)}{\left(N + \sum_{j \in \Omega} y_j(1 - \hat{y}_j)\right)^2}.
\end{aligned}
$$

We may observe the following from the derived expression:

1. The gradient is proportional to $y_i$:

$$
\frac{\partial \mathscr{L}_{\mathrm{ESL}}}{\partial \hat{y}_i} \propto y_i.
$$

   Therefore, if $y_i = 0$ (corresponding to a negative pixel), then

$$
\frac{\partial \mathscr{L}_{\mathrm{ESL}}}{\partial \hat{y}_i} = 0.
$$

   This shows that True Negatives (TN) and FP pixels do not contribute to the gradient, and the loss specifically emphasizes the positive pixels ($y_i = 1$), i.e., the FN regions.

2. The numerator term $(N + \sum_{j \in \Omega} y_j)$ is constant for a given image. Only the denominator

$$
D = N + \sum_{j \in \Omega} y_j(1 - \hat{y}_j)
$$

   varies with the predicted values, and it decreases as the number of correctly predicted positive pixels increases. Since

$$
\sum_{j \in \Omega} y_j(1 - \hat{y}_j) \le \sum_{j \in \Omega} y_j,
$$

   the denominator is always bounded below by $N$, preventing the gradient magnitude from becoming excessively large.

Thus, including $N$ in the denominator ensures numerical stability:

$$
\left|\frac{\partial \mathscr{L}_{\mathrm{ESL}}}{\partial \hat{y}_i}\right| = \frac{y_i \left(N + \sum_{j \in \Omega} y_j\right)}{\left(N + \sum_{j \in \Omega} y_j(1 - \hat{y}_j)\right)^2} \le \frac{y_i \left(N + \sum_{j \in \Omega} y_j\right)}{N^2} \le \frac{y_i N}{N^2},
$$

$$
\implies \left|\frac{\partial \mathscr{L}_{\mathrm{ESL}}}{\partial \hat{y}_i}\right| \le \frac{y_i}{N}.
$$

Since $y_i \in \{0, 1\}$, we have

$$
\left|\frac{\partial \mathscr{L}_{\mathrm{ESL}}}{\partial \hat{y}_i}\right| \le \frac{1}{N}.
$$

,

which guarantees bounded and stable gradients even for sparse positive targets, ensuring stable optimization throughout training.

This analysis highlights that the gradient flow for our loss function is entirely concentrated on positive pixels, directly targeting the FN regions while ignoring TN and FP contributions. Moreover, the presence of $N$ in the denominator effectively scales the gradient, ensuring that its magnitude remains bounded by $1/N$ regardless of the number of positive pixels or their predictions. Consequently, the loss maintains sensitivity to challenging regions without causing unstable or excessively large updates, supporting consistent and stable training even for sparse masks. Thus, $N$ acts as a normalization factor, balancing sensitivity to false negatives with overall numerical stability.

## D    TESTING SMM ON SEGNET

To further demonstrate the versatility and robustness of our proposed framework, we evaluated both variants of SMM on SegNet (Badrinarayanan et al., 2017), a widely used segmentation architecture that differs from U-Net in its encoder-decoder design and feature propagation strategy. This experiment highlights the architecture-agnostic nature of our approach, showing that the unified mask modulation and generalizable training strategy can be applied to diverse segmentation networks while maintaining high performance. Table 4 presents the test set metrics for SegNet trained under

Table 4: **Test set metrics** of SegNet models trained using different strategies. SMM*v1* and SMM*v2* present results for both versions of our proposed framework.

| Method | DSC ↑ | clDice ↑ | JSI ↑ | FNR ↓ | FPR ↓ |
|---|---|---|---|---|---|
| **BoMBR** (Raina et al., 2024) | | | | | |
| SegNet | $64.35 \pm 1.82$ | $61.56 \pm 1.97$ | $54.26 \pm 2.05$ | $26.11 \pm 2.02$ | $8.17 \pm 0.65$ |
| SegNet + SRL | $64.67 \pm 2.89$ | $62.33 \pm 3.05$ | $54.83 \pm 2.89$ | $27.00 \pm 3.09$ | $8.08 \pm 1.08$ |
| SegNet + BL | $64.12 \pm 2.00$ | $61.94 \pm 1.42$ | $54.86 \pm 2.16$ | $28.16 \pm 0.72$ | $8.43 \pm 1.07$ |
| SMM*v1* | $\mathbf{66.52 \pm 1.07}$ | $\mathbf{64.17 \pm 1.21}$ | $\mathbf{56.99 \pm 1.19}$ | $\mathbf{25.62 \pm 1.72}$ | $\mathbf{7.33 \pm 0.20}$ |
| SMM*v2* | $65.76 \pm 1.02$ | $63.87 \pm 0.93$ | $56.31 \pm 1.14$ | $25.92 \pm 1.21$ | $7.83 \pm 0.83$ |
| **DRIVE** (Hassan et al., 2015) | | | | | |
| SegNet | $66.52 \pm 5.41$ | $66.32 \pm 5.61$ | $50.13 \pm 5.94$ | $38.07 \pm 5.03$ | $3.33 \pm 0.71$ |
| SegNet + SRL | $63.96 \pm 3.10$ | $63.70 \pm 3.04$ | $47.17 \pm 3.42$ | $38.28 \pm 3.15$ | $4.30 \pm 0.46$ |
| SegNet + BL | $65.51 \pm 4.90$ | $66.10 \pm 5.27$ | $48.99 \pm 5.55$ | $42.83 \pm 6.07$ | $\mathbf{2.32 \pm 0.49}$ |
| SMM*v1* | $66.63 \pm 4.92$ | $66.02 \pm 4.95$ | $50.23 \pm 5.44$ | $38.54 \pm 5.10$ | $3.14 \pm 0.50$ |
| SMM*v2* | $\mathbf{67.06 \pm 4.96}$ | $\mathbf{66.79 \pm 5.06}$ | $\mathbf{50.72 \pm 5.50}$ | $\mathbf{36.22 \pm 5.06}$ | $3.63 \pm 0.62$ |

various strategies, including baseline training, self-regularized learning (SRL), boundary loss (BL), and our proposed SMM variants. Across both datasets, SMM consistently improves segmentation performance compared to standard training strategies, achieving higher Dice, clDice, and Jaccard scores while reducing false negative and false positive rates. These results confirm that the effectiveness of SMM is not confined to a single architecture, underscoring its potential for broad deployment across different segmentation models.

## E    VERSIONAL DESIGN LED SUPERIORITY

The results presented in Table 1 highlight the distinct behaviors of the two variants of SMM across datasets with different characteristics. For clarity, we classify the datasets into two categories:

1. **Negative-dominant datasets**: The background is considerably more diverse and substantially larger than the foreground. Segmentation in such cases is particularly challenging due to bias toward the more abundant negative class. Representative datasets include DRIVE (Hassan et al., 2015) and Cracks (Tomaszkiewicz & Owerko, 2023).

2. **Balanced or foreground-rich datasets**: The classes are approximately balanced, or foreground pixels slightly dominate. Here, the primary challenge lies in capturing fine struc-

tural details and ensuring accurate delineation of class boundaries. Examples include BoMBR (Raina et al., 2024) and Drone[1].

For architectures such as U-Net, SMM*v1* consistently improves overlap- and topology-oriented metrics relative to vanilla baselines, with the largest gains observed in Category 1 datasets. This observation emphasizes the effectiveness of the ESL loss in tasks dominated by negative samples, thereby positioning SMM*v1* as the more robust variant under such conditions. In contrast, SMM*v2* demonstrates superior performance on Category 2 datasets, reflecting its suitability for scenarios where the segmentation task depends less on class imbalance and more on semantic precision and fine-grained contextual reasoning. These findings are further substantiated by the qualitative results shown in Figure 3. It may be noticed that in cases of Category 1 datasets, SMM*v1* is able to segregate the regions missed by the baselines. SMM*v2*, on the other hand, efficiently balances under-prediction and over-prediction in the case of Category 2 datasets, thus ensuring accurate semantic segregation of separate classes.

*It should be noted that the trends mentioned above are observed in architectures that are not specifically tailored to a particular domain or task (e.g., U-Net). We do not assert these patterns as universal across all segmentation models.*

Interestingly, the trend reverses when considering the results in Table 4. In this case, SMM*v1* exhibits stronger performance on Category 2, whereas SMM*v2* proves more effective for Category 1 datasets.