# OpenReview forum: "Balancing the False Positive-Negative tradeoff to enhance Image Segmentation"
_ICLR.cc/2026/Conference — Submitted to ICLR 2026_

### Official Review · Reviewer_rVMb · 2025-10-30

**Soundness:** 2
**Presentation:** 2
**Contribution:** 2
**Rating:** 2
**Confidence:** 5

**Summary:**

(1) Exploits the hypothesis that the number of FN is significantly higher than the number of FP. We attempt to improve the model’s performance by introducing intended FP, conditioned by model performance, into the ground truth masks for enhanced training, thereby penalizing the model for missing out class pixels in smaller regions or some structures entirely. The results validate that this strategy tends to bring an overall improvement in the model performance.
(2) Proved effective across diverse datasets, demonstrating improved performance on both binary and multi-class segmentation tasks. Its versatility makes it applicable to a wide range of imaging scenarios

**Strengths:**

(1) Authors evaluated their method on difference tasks and datasets.

(2) Quantitative and qualitative results were provided to help readers understand the benefits of the proposed methods.

**Weaknesses:**

(1) The overall contribution is low. The proposed method did not improve the segmentation performance significantly. In contrast, the proposed method demonstrated a similar performance with U-Net (67.46 vs. 67.09 and 80.64 vs 80.01). Thus, if this method was proposed with other more advanced methods instead of U-Net, it would not outperform them.

(2) The overall novelty is low. Authors only proposed a Miss-aware Mask Modulation which included a simple structure. Based on the segmentation results reported in different tasks, this proposed module did not demonstrate significant improvements over existing methods.

(3) The reproducibility is low. Some implementation details are missing, and authors did not provide these details.

(4) Author did not provide computational complexity of the proposed method. Authors added a new module into the baseline network, such as U-Net, so it is necessary to show the increased parameters and FLOPs due to the incorporation of this module.

(5) The evaluation is insufficient. Authors only evaluated their method on convolutional neural networks, but they did not evaluate it on Vision transformer-based segmentation models.

**Questions:**

(1) The overall contribution is low. The proposed method did not improve the segmentation performance significantly. In contrast, the proposed method demonstrated a similar performance with U-Net (67.46 vs. 67.09 and 80.64 vs 80.01). Thus, if this method was proposed with other more advanced methods instead of U-Net, it would not outperform them.

(2) The overall novelty is low. Authors only proposed a Miss-aware Mask Modulation which included a simple structure. Based on the segmentation results reported in different tasks, this proposed module did not demonstrate significant improvements over existing methods.

(3) The reproducibility is low. Some implementation details are missing, and authors did not provide these details.

(4) Author did not provide computational complexity of the proposed method. Authors added a new module into the baseline network, such as U-Net, so it is necessary to show the increased parameters and FLOPs due to the incorporation of this module.

(5) The evaluation is insufficient. Authors only evaluated their method on convolutional neural networks, but they did not evaluate it on Vision transformer-based segmentation models.

---

> ### Author Response · Authors · 2025-11-23
> **Clarifications on Statistical Significance, Reproducibility, and Methodological Design**
>
> Dear Reviewer,
>
> Thank you for your valuable insights on our work. We present our responses below in the order of their appearance.
>
> **Statistical Significance**: We evaluated our metrics using Cohen’s d effect size. Across several datasets and metrics, our models show moderate positive effect sizes relative to the strongest baseline, indicating that the improvements are systematic rather than incidental. While the results are not statistically significant under our current setup of five seeds, the observed effect sizes suggest that increasing the number of test cases would enhance statistical power and likely make these differences significant. We will include this discussion in the updated manuscript.
>
> Note: We also evaluated Tversky and Focal losses as additional baselines, and our method outperforms them across most metrics for all datasets. The table below includes these baselines. The t-value has been calculated using the Welch t-test.
> | Dataset | Metric | t-value | p-value | Effect Size (d) |
> |--------|--------|---------|---------|------------------|
> | **BoMBR** | DSC | 0.507 | 0.313 | 0.321 |
> |          | clDice | -2.742 | 0.013 | -1.734 |
> |          | JSI | 0.233 | 0.411 | 0.147 |
> |          | FNR | -0.994 | 0.179 | -0.628 |
> |          | FPR | -0.575 | 0.291 | -0.364 |
> | **DRIVE** | DSC | 0.600 | 0.284 | 0.380 |
> |          | clDice | 0.196 | 0.426 | 0.124 |
> |          | JSI | 0.610 | 0.281 | 0.386 |
> |          | FNR | 2.075 | 0.043 | 1.313 |
> |          | FPR | 0.906 | 0.197 | 0.573 |
> | **Cracks** | DSC | 0.426 | 0.345 | 0.269 |
> |          | clDice | 0.802 | 0.231 | 0.507 |
> |          | JSI | 0.649 | 0.274 | 0.410 |
> |          | FNR | 3.327 | 0.006 | 2.104 |
> |          | FPR | 3.162 | 0.007 | 2.000 |
> | **Drone** | DSC | 1.129 | 0.146 | 0.714 |
> |          | clDice | 1.258 | 0.122 | 0.796 |
> |          | JSI | 1.043 | 0.164 | 0.659 |
> |          | FNR | -1.121 | 0.148 | -0.709 |
> |          | FPR | -1.018 | 0.169 | -0.644 |
>
> **Reproducibility**: Sections 5.2 and 5.3 provide the model architectures and hyperparameters required for replication. We used standard dataset splits. Following double-blind requirements, we did not provide the GitHub link. We will include the full implementation link in the final version.
>
> **Computational Complexity**: Our method introduces no new trainable parameters, so the computational complexity depends entirely on the model architecture. The extra cost comes only from lightweight algebraic operations for mask modulation and loss computation, adding negligible overhead relative to the forward and backward passes.
>
> **Baseline Architectures**: We reported results on two widely used segmentation baselines. The consistent improvements indicate broad applicability. Based on your suggestion, we are also evaluating our methods on transformer-based architectures. Since they are computationally heavy, we request some additional time to finalize the results. They will be included in the updated manuscript.
>
> **Additional**: We further compared against Tversky and Focal Loss on U-Net. The improvements reinforce the robustness of our approach.
> | Method | DSC ↑ | clDice ↑ | JSI ↑ | FNR ↓ | FPR ↓ |
> | :--- | :---: | :---: | :---: | :---: | :---: |
> | **BoMBR** | | | | | |
> | Tversky | 66.64 ± 0.96 | 63.76 ± 0.39 | 57.18 ± 1.02 | 25.87 ± 1.25 | 7.37 ± 0.25 |
> | Focal | 65.96 ± 1.14 | **65.53 ± 0.64** | 56.37 ± 1.17 | 26.05 ± 1.32 | 7.49 ± 0.64 |
> | SMM*v1* | 66.82 ± 1.16 | 64.12 ± 0.95 | 57.37 ± 1.38 | 25.92 ± 0.77 | 7.27 ± 0.39 |
> | SMM*v2* | **67.46 ± 1.24** | 64.42 ± 0.64 | **57.96 ± 1.13** | **24.73 ± 1.14** | **7.09 ± 0.39** |
> | **DRIVE** | | | | | |
> | Tversky | 79.79 ± 0.34 | 83.70 ± 0.75 | 66.41 ± 0.48 | 21.26 ± 0.52 | 2.55 ± 0.11 |
> | Focal | 80.22 ± 0.87 | 84.24 ± 0.88 | 67.02 ± 1.22 | **20.84 ± 1.03** | 2.48 ± 0.09 |
> | SMM*v1* | **80.64 ± 1.30** | **84.42 ± 1.51** | **67.62 ± 1.83** | 20.98 ± 2.08 | **2.31 ± 0.26** |
> | SMM*v2* | 78.93 ± 0.68 | 82.71 ± 0.94 | 65.24 ± 0.92 | 21.53 ± 1.08 | 2.79 ± 0.09 |
> | **Cracks** | | | | | |
> | Tversky | 64.33 ± 0.36 | 74.75 ± 0.35 | 51.02 ± 0.24 | **30.97 ± 0.50** | 0.34 ± 0.00|
> | Focal | 62.10 ± 3.13 | 71.15 ± 4.93 | 48.91 ± 3.25 | 33.18 ± 3.97 | 0.40 ± 0.19 |
> | SMM*v1* | **64.74 ± 0.20** | **75.35 ± 0.34** | **51.44 ± 0.21** | 31.16 ± 0.58 | **0.33 ± 0.01** |
> | SMM*v2* | 62.93 ± 2.73 | 72.64 ± 4.50 | 49.56 ± 2.80 | 33.08 ± 2.82 | **0.33 ± 0.02** |
> | **Drone** | | | | | |
> | Tversky | 49.64 ± 2.37 | 44.49 ± 2.23 | 40.23 ± 1.85 | 30.54 ± 3.34 | 6.13 ± 0.29 |
> | Focal | 47.19 ± 2.99 | 42.47 ± 3.18 | 38.08 ± 2.37 | 32.59 ± 3.33 | 6.44 ± 0.25 |
> | SMM*v1* | 50.49 ± 1.72 | 45.45 ± 1.91 | 40.89 ± 1.46 | 29.20 ± 1.79 | 6.07 ± 0.28 |
> | SMM*v2* | **51.34 ± 2.39** | **46.21 ± 2.09** | **41.61 ± 2.31** | **27.70 ± 2.20** | **5.93 ± 0.33** |
>
> We hope these points clarify the concerns raised in the comment. We are happy to provide any additional information that may be needed to clarify our method. We will include the suggested updates in the final version of the paper.

---

### Official Review · Reviewer_as4E · 2025-10-31

**Soundness:** 3
**Presentation:** 3
**Contribution:** 2
**Rating:** 2
**Confidence:** 4

**Summary:**

This work is motivated by the observation that, for medical segmentation tasks, false positives and false negatives are usually not well balanced, resulting in a disproportionate emphasis on one of them.
 The authors proposed a supervised mask modulation (SMM) method to address this issue. The proposed method is architecture-agnostic. SMM improves performance by introducing intentional false positives, under the hypothesis that, for certain tasks, the false negative rate (FNR) is higher than the false positive rate (FPR) due to the small segmentation region (class imbalance).
 The proposed method was tested on four datasets with two variants.

**Strengths:**

1. The motivation of this work is clearly articulated, and I appreciate the effort to address the observed issues, especially in the medical domain, where further research is certainly needed.
2. The authors run multiple seeds for performance evaluation and conduct proper statistical analyses. This aspect is often overlooked in machine learning research, even though it should not be. I appreciate the authors’ efforts in ensuring the reproducibility of this work.
3. The presentation is clear and easy to follow, with figures that are both intuitive and informative.

**Weaknesses:**

1. The proposed solution appears to be quite hard-coded. To my understanding, the core idea of the method is to force the model to learn a larger mask through dilation. This raises a concern: what if the class imbalance is reversed, i.e., there are more samples of class = 1 than class = 0? In that case, one would likely need to invert the strategy by shrinking the mask instead. This makes the proposed method heavily hard-coded. It would be more interesting if the authors could propose an adaptive mechanism that automatically accounts for the degree and direction of class imbalance. In this regard, I believe there is substantial room for improvement.

2. The structure of the paper could also be improved. It seems that there are two main components of related work or background information that deserve emphasis. The first concerns evidence from prior work showing that the false negative rate (FNR) tends to exceed the false positive rate (FPR), and the second concerns previous efforts to address this issue and how the proposed method differs from them. Currently, the first part appears in Section 3, and the second is in the first paragraph of Section 2. Both aspects are also briefly mentioned in the introduction but without any citations. I recommend that the authors: (1) cite relevant literature whenever the issue or existing solutions are discussed, and (2) consolidate these contents into a dedicated section with clear subsections.

3. Finally, there is a lack of discussion regarding the results. For example, in Figure 4, it is unclear what the yellow arrow represents and how it should be interpreted. The caption is not sufficiently intuitive. I would suggest saving some space from, for instance, Figure 3 (which could be made smaller or even omitted) and using it to include more paragraphs analyzing and discussing the results in greater depth.

**Questions:**

1. There’s no weighting inbetween the 2 loss terms in algorithm 2?
2. Isn’t that the Loss of ESL, is: $L_{ESL} = - \frac{TP+FP}{N+FP+FN}$? I am not sure why this could help the better balance between FP and FN as they have the same weights (weights = 1.0 for both of them). Also do you have any intuition behind this design?

---

> ### Author Response · Authors · 2025-11-23
> **Explanations on Design Decisions, Multi-Class Handling, and Benchmark Comparisons**
>
> Dear Reviewer,
> Thank you for your valuable insights on our work. We present our responses below in the order of their occurrence.
>
> **Weight factor**: We chose not to add a weight factor because [1] shows that adding weight does not lead to significant improvement, as any value of weight leads to convergence at a similar model configuration asymptotically.
>
> **Mathematical Analysis of ESL**: ESL uses TP/(N+FN). This sensitivity-based loss is designed to reduce False Negatives by ensuring that the gradient flow focuses entirely on positive pixels, targeting FN regions while ignoring TN and FP. The mathematical details, including the role of N, are discussed in Appendix B.
>
> We further address the highlighted weakness of the paper:
>
> **Adaptive Methodology**: Our mask modulation is adaptive to the degree and direction of missing pixels. Miss Aware Mask Modulation (MAMM) introduces false negatives wherever the model prediction misses a class. For multi-class settings, channel-wise masks ensure dilation in one class does not affect another. This results in a multi-label setup; SMMv2 uses classwise BCE to handle overlaps, while SMMv1 keeps vanilla losses on original masks and applies modulation only within ESL.
>
> **Manuscript Format**: Thank you for this suggestion. We will revise the format in our updated submission.
>
> **Caption Clarity**: We will incorporate more detailed caption explanations. The yellow arrows highlight areas showing notable differences in the model outputs.
>
> To further strengthen our method, we evaluated it against additional FNR-based baselines (Tversky and Focal Loss). Our methods show improved performance compared to these baselines as well, reinforcing their robustness across diverse state-of-the-art baselines.
>
> | Method | DSC ↑ | clDice ↑ | JSI ↑ | FNR ↓ | FPR ↓ |
> | :--- | :---: | :---: | :---: | :---: | :---: |
> | **BoMBR** | | | | | |
> | Vanilla U-Net | 66.02 ± 2.11 | 63.57 ± 1.76 | 56.42 ± 2.44 | 26.04 ± 1.82 | 7.58 ± 0.81 |
> | U-Net + SRL | 66.84 ± 2.21 | 63.54 ± 1.36 | 57.24 ± 2.41 | 25.78 ± 2.07 | 7.29 ± 0.95 |
> | U-Net + BL | 67.09 ± 1.06 | 64.15 ± 1.29 | 57.80 ± 1.04 | 26.11 ± 1.15 | 7.23 ± 0.38 |
> | U-Net + Tversky | 66.64 ± 0.96 | 63.76 ± 0.39 | 57.18 ± 1.02 | 25.87 ± 1.25 | 7.37 ± 0.25 |
> | U-Net + Focal | 65.96 ± 1.14 | **65.53 ± 0.64** | 56.37 ± 1.17 | 26.05 ± 1.32 | 7.49 ± 0.64 |
> | SMM*v1* | 66.82 ± 1.16 | 64.12 ± 0.95 | 57.37 ± 1.38 | 25.92 ± 0.77 | 7.27 ± 0.39 |
> | SMM*v2* | **67.46 ± 1.24** | 64.42 ± 0.64 | **57.96 ± 1.13** | **24.73 ± 1.14** | **7.09 ± 0.39** |
> | **DRIVE** | | | | | |
> | Vanilla U-Net | 79.63 ± 1.45 | 83.48 ± 1.84 | 66.21 ± 1.98 | 21.51 ± 2.00 | 2.55 ± 0.10 |
> | U-Net + SRL | 80.01 ± 0.47 | 84.27 ± 0.81 | 66.72 ± 0.65 | **18.85 ± 0.97** | 2.97 ± 0.08 |
> | U-Net + BL | 79.72 ± 1.74 | 83.36 ± 1.48 | 66.35 ± 2.39 | 23.40 ± 1.13 | **2.12 ± 0.39** |
> | U-Net + Tversky | 79.79 ± 0.34 | 83.70 ± 0.75 | 66.41 ± 0.48 | 21.26 ± 0.52 | 2.55 ± 0.11 |
> | U-Net + Focal | 80.22 ± 0.87 | 84.24 ± 0.88 | 67.02 ± 1.22 | 20.84 ± 1.03 | 2.48 ± 0.09 |
> | SMM*v1* | **80.64 ± 1.30** | **84.42 ± 1.51** | **67.62 ± 1.83** | 20.98 ± 2.08 | 2.31 ± 0.26 |
> | SMM*v2* | 78.93 ± 0.68 | 82.71 ± 0.94 | 65.24 ± 0.92 | 21.53 ± 1.08 | 2.79 ± 0.09 |
> | **Cracks** | | | | | |
> | Vanilla U-Net | 64.57 ± 0.87 | 74.92 ± 1.15 | 51.20 ± 0.80 | 31.39 ± 1.22 | 0.33 ± 0.01 |
> | U-Net + SRL | 62.51 ± 3.31 | 71.93 ± 5.15 | 49.12 ± 3.36 | **29.69 ± 0.80** | 0.44 ± 0.13 |
> | U-Net + BL | 64.05 ± 0.93 | 74.73 ± 0.82 | 50.82 ± 0.93 | 33.15 ± 0.66 | **0.31 ± 0.01** |
> | U-Net + Tversky | 64.33 ± 0.36 | 74.75 ± 0.35 | 51.02 ± 0.24 | 30.97 ± 0.50 | 0.34 ± 0.00 |
> | U-Net + Focal | 62.10 ± 3.13 | 71.15 ± 4.93 | 48.91 ± 3.25 | 33.18 ± 3.97 | 0.40 ± 0.19 |
> | SMM*v1* | **64.74 ± 0.20** | **75.35 ± 0.34** | **51.44 ± 0.21** | 31.16 ± 0.58 | 0.33 ± 0.01 |
> | SMM*v2* | 62.93 ± 2.73 | 72.64 ± 4.50 | 49.56 ± 2.80 | 33.08 ± 2.82 | 0.33 ± 0.02 |
> | **Drone** | | | | | |
> | Vanilla U-Net | 49.58 ± 2.54 | 44.44 ± 2.34 | 39.95 ± 2.04 | 29.47 ± 2.76 | 6.32 ± 0.25 |
> | U-Net + SRL | 48.92 ± 1.45 | 43.92 ± 1.38 | 39.16 ± 1.02 | 29.69 ± 1.23 | 6.40 ± 0.24 |
> | U-Net + BL | 45.37 ± 8.06 | 40.13 ± 7.81 | 37.76 ± 6.60 | 36.23 ± 7.17 | 6.73 ± 1.28 |
> | U-Net + Tversky | 49.64 ± 2.37 | 44.49 ± 2.23 | 40.23 ± 1.85 | 30.54 ± 3.34 | 6.13 ± 0.29 |
> | U-Net + Focal | 47.19 ± 2.99 | 42.47 ± 3.18 | 38.08 ± 2.37 | 32.59 ± 3.33 | 6.44 ± 0.25 |
> | SMM*v1* | 50.49 ± 1.72 | 45.45 ± 1.91 | 40.89 ± 1.46 | 29.20 ± 1.79 | 6.07 ± 0.28 |
> | SMM*v2* | **51.34 ± 2.39** | **46.21 ± 2.09** | **41.61 ± 2.31** | **27.70 ± 2.20** | **5.93 ± 0.33** |
>
> We hope these points address your concerns. We are happy to provide any additional clarification and will incorporate the suggested updates in the final version of the manuscript.
>
>
>
> [1] Jonathon Byrd and Zachary Lipton. What is the effect of importance weighting in deep learning? International Conference on Machine Learning, pages 872–881. PMLR, 2019.

---

> ### Comment · Reviewer_as4E · 2025-11-24
>
> Thanks for the reply. I don't really buy the most of the responses so I will keep my score. Here are the details:
> 1. weight factor. The paper you listed is also an empirical paper (not theoretically proof that this apply to all situation) which conducted on different task with different datasets (CIFAR 10 binary classification) -- I dont find it a very powerful support on why you choose this kind of design with your task (segmentation for medical imaging).
> 2. ESL loss. Eq 1 in your paper (line 216 - 219). lets take a look of the numerator. Isnt that when y_i and \hat{y_i} both == 0, the product of them will be 1 and this adds up to the sum? how can the numerator be only the TP but not TP+TN? same for denominator.
>
> --
> 3. I don't think the authors answers my questions at all. My question was, the method seems to be hard-coded, which only make sense when FP is more than FN, as the method seems to force the model to learn a bigger mask with dilation. How about when the situation is different that FN is more than FP? I don't think the authors response to that.
> 4/5. I didn't find the update in the current version. This is minors compared to the rest -- the questions and the weakness mentioned before are not well addressed in my opinion..
>
> About the extra experiment. I didn't ask for it, and I don't think those answered any of my questions. Besides of that, I don't think that's fair to provide extra experiments without a specific request from reviewers (the deadline for all submissions should be the same). Maybe other reviewers asked that but I didn't. so I would prob simply ignore this part.
>
> In general, the responses do not answer my questions.. so I kept the same score.

---

> > ### Author Response · Authors · 2025-11-26
> > **Explanations on weight factor, ESL equation and robustness of approach**
> >
> > Dear Reviewer,
> > Thank you for your valuable insights on our work. We present our responses below in the order of their occurrence.
> >
> > **Weight factor**: Incorporating a weight factor is a minor task, and we can certainly consider including an analysis of it in the final version of the paper, subject to time constraints. However, even in the current formulation, it can be inferred that we set the weighting factor for the loss to 1 for all our experiments, and practitioners may adjust or tune this parameter as needed for their specific use cases.
> >
> > **Mathematical Analysis of ESL**: The loss uses plain multiplication and addition. In Eq. 1, the numerator contains the term $\(y_i \cdot \hat{y}_i\)$. Although both inputs are binary, this is numeric multiplication. Under numeric multiplication, the product is 1 only when both terms are 1. When both are 0, the product is $\(0 \cdot 0 = 0\)$, not 1. Thus, the numerator accumulates only TP, not TN. Similarly, the denominator term $\((1 - \hat{y}_i) \cdot y_i\)$ evaluates to 1 only for $\(y_i = 1, \hat{y}_i = 0\)$, i.e., FN. When both are 0, the product is $\(1 \cdot 0 = 0\)$. Therefore, TNs do not enter the denominator either. The core point is that the formulation uses numeric products of binary values, not Boolean logic, so cases like $\(y_i = 0, \hat{y}_i = 0\)$ do not produce a value of 1 anywhere in the loss.
> >
> > **Case of High False Negatives**: This scenario is highly improbable. While we do not currently provide a formal theoretical proof, our hypothesis is supported by strong empirical evidence.
> > Moreover, in the initial epochs, the model may surely exhibit a higher number of false positives; but they are not included in the mask and simply ignored and so the mask modulation strategy yields a mask equivalent to the ground truth if a mask with significantly high FPR ever shows up. False positives are typically easier for the model to suppress, and as training progresses, they become significantly fewer than the false negatives.
> > Even in the unlikely event that this does not occur, the model is not constrained in any way and simply trains on the ground truth masks. In practice, we have not encountered any case where the FPR even comes close, let alone exceeds, the FNR.
> >
> > We hope these points address your concerns. We are happy to provide any additional clarification and will incorporate the suggested updates in the final version of the manuscript.

---

### Official Review · Reviewer_oQze · 2025-10-31

**Soundness:** 2
**Presentation:** 3
**Contribution:** 2
**Rating:** 2
**Confidence:** 4

**Summary:**

The paper introduces Supervised Mask Modulation (SMM), a training-time, architecture-agnostic approach for image segmentation that aims to reduce false negatives while controlling false positives. The core idea (MAMM) dilates predicted FN regions and merges them into the ground truth masks during training. Two variants are presented: SMMv1 (with an elevated-sensitivity loss) and SMMv2 (with an adaptive threshold guided by recall trends). Experiments span several datasets (BoMBR, DRIVE, Cracks, Drone) with multiple metrics.

**Strengths:**

1. Problem motivation is clear and relevant, especially in FN-sensitive domains (e.g., medical, defects).

2. The method integrates at training time without architectural changes, which is easy to adopt.

3. Experiments cover multiple datasets and metrics.

4. The paper is generally readable.

**Weaknesses:**

1. Novelty: The mechanism—dilating FN regions and augmenting labels—resembles label modulation/cost-sensitive training; ESL aligns with recall-weighted objectives. Without a clearer theoretical account or principled links to established losses, the contribution feels incremental for ICLR.

2. Baselines: Key baselines are missing (Tversky, Focal), and evaluations focus on U-Net (and SegNet in the appendix) without modern backbones (e.g., DeepLabv3+, transformer-based segmenters), limiting the architecture-agnostic claim.

3. Dataset: The datasets are small and 2D-only, which limits assessment of scalability and modern applicability. If a 2D focus is intentional, the paper should justify this choice and discuss applicability to volumetric/3D tasks.

4. Result: The main text makes a strong claim that the approach has been "validated on a range of benchmark datasets, consistently outperforming state-of-the-art methods," yet Appendix A shows most improvements are not statistically significant. The authors' own summary acknowledges that "while not all improvements reach statistical significance, there are multiple encouraging trends in favor of our approach."

**Questions:**

1. Why restrict evaluation to 2D tasks when many target domains use volumetric/3D data? Can SMM be extended to volumetric segmentation, and what limitations would arise?

2. How should readers reconcile the strong claim in the main text with Appendix A showing limited significance (e.g., BoMBR/DRIVE: all p>0.05; Cracks: clDice/JSI; Drone: FPR)? Can you report effect sizes and confidence intervals?

3. How does SMM compare with tuned Tversky and Focal losses? Where does SMM provide unique benefits?

4. How do you ensure modulated masks remain bounded and do not overfit to early noise? How are overlaps handled in multi-class settings in practice?

5. What is the training time and memory overhead relative to the baseline pipeline?

---

> ### Author Response · Authors · 2025-11-23
> **Clarifications on Significance Analysis, Baseline Comparisons, and Methodological Design**
>
> Dear Reviewer,
> Thank you for your valuable insights on our work. We present our responses below in the order of your comments.
>
> **Volumetric Tests**: The methodology naturally extends to volumetric scans, but we could not run these experiments due to computational constraints. Nonetheless, the efficiency demonstrated by our method indicates consistent performance gains across diverse tasks and across state-of-the-art base models, thereby justifying its applicability.
>
> **Effect Sizes**: As suggested, we computed Cohen’s *d* for all datasets and metrics. Our models show moderate positive effect sizes over the strongest baseline, indicating systematic rather than incidental improvements. While the results are not statistically significant with only five seeds, the observed effect sizes suggest that increasing the number of test cases would enhance statistical power and likely yield significance.
>
> Note: These metrics have been derived by comparing the best-performing methodology (SMM*v1* or SMM*v2*) with the best baseline. *The baselines now also include Focal and Tversky Loss (as highlighted in the next point)*.
> | Dataset | Metric | t-value | p-value | Effect Size (d) |
> |--------|--------|---------|---------|------------------|
> | **BoMBR** | DSC | 0.507 | 0.313 | 0.321 |
> | | clDice | -2.742 | 0.013 | -1.734 |
> | | JSI | 0.233 | 0.411 | 0.147 |
> | | FNR | -0.994 | 0.179 | -0.628 |
> | | FPR | -0.575 | 0.291 | -0.364 |
> | **DRIVE** | DSC | 0.600 | 0.284 | 0.380 |
> | | clDice | 0.196 | 0.426 | 0.124 |
> | | JSI | 0.610 | 0.281 | 0.386 |
> | | FNR | 2.075 | 0.043 | 1.313 |
> | | FPR | 0.906 | 0.197 | 0.573 |
> | **Cracks** | DSC | 0.426 | 0.345 | 0.269 |
> | | clDice | 0.802 | 0.231 | 0.507 |
> | | JSI | 0.649 | 0.274 | 0.410 |
> | | FNR | 3.327 | 0.006 | 2.104 |
> | | FPR | 3.162 | 0.007 | 2.000 |
> | **Drone** | DSC | 1.129 | 0.146 | 0.714 |
> | | clDice | 1.258 | 0.122 | 0.796 |
> | | JSI | 1.043 | 0.164 | 0.659 |
> | | FNR | -1.121 | 0.148 | -0.709 |
> | | FPR | -1.018 | 0.169 | -0.644 |
>
> **Better Baselines**: Following your suggestion, we added comparisons with Tversky and Focal losses on **U-Net**. The results reaffirm the effectiveness of our method, which consistently outperforms these baselines on most metrics across all datasets.
>
> (**Best values in each column are bolded**.)
>
> | Method | DSC ↑ | clDice ↑ | JSI ↑ | FNR ↓ | FPR ↓ |
> | :--- | :---: | :---: | :---: | :---: | :---: |
> | **BoMBR** | | | | | |
> | Tversky | 66.64 ± 0.96 | 63.76 ± 0.39 | 57.18 ± 1.02 | 25.87 ± 1.25 | 7.37 ± 0.25 |
> | Focal | 65.96 ± 1.14 | **65.53 ± 0.64** | 56.37 ± 1.17 | 26.05 ± 1.32 | 7.49 ± 0.64 |
> | SMM*v1* | 66.82 ± 1.16 | 64.12 ± 0.95 | 57.37 ± 1.38 | 25.92 ± 0.77 | 7.27 ± 0.39 |
> | SMM*v2* | **67.46 ± 1.24** | 64.42 ± 0.64 | **57.96 ± 1.13** | **24.73 ± 1.14** | **7.09 ± 0.39** |
> | **DRIVE** | | | | | |
> | Tversky | 79.79 ± 0.34 | 83.70 ± 0.75 | 66.41 ± 0.48 | 21.26 ± 0.52 | 2.55 ± 0.11 |
> | Focal | 80.22 ± 0.87 | 84.24 ± 0.88 | 67.02 ± 1.22 | **20.84 ± 1.03** | 2.48 ± 0.09 |
> | SMM*v1* | **80.64 ± 1.30** | **84.42 ± 1.51** | **67.62 ± 1.83** | 20.98 ± 2.08 | **2.31 ± 0.26** |
> | SMM*v2* | 78.93 ± 0.68 | 82.71 ± 0.94 | 65.24 ± 0.92 | 21.53 ± 1.08 | 2.79 ± 0.09 |
> | **Cracks** | | | | | |
> | Tversky | 64.33 ± 0.36 | 74.75 ± 0.35 | 51.02 ± 0.24 | **30.97 ± 0.50** | 0.34 ± 0.00|
> | Focal | 62.10 ± 3.13 | 71.15 ± 4.93 | 48.91 ± 3.25 | 33.18 ± 3.97 | 0.40 ± 0.19 |
> | SMM*v1* | **64.74 ± 0.20** | **75.35 ± 0.34** | **51.44 ± 0.21** | 31.16 ± 0.58 | **0.33 ± 0.01** |
> | SMM*v2* | 62.93 ± 2.73 | 72.64 ± 4.50 | 49.56 ± 2.80 | 33.08 ± 2.82 | **0.33 ± 0.02** |
> | **Drone** | | | | | |
> | Tversky | 49.64 ± 2.37 | 44.49 ± 2.23 | 40.23 ± 1.85 | 30.54 ± 3.34 | 6.13 ± 0.29 |
> | Focal | 47.19 ± 2.99 | 42.47 ± 3.18 | 38.08 ± 2.37 | 32.59 ± 3.33 | 6.44 ± 0.25 |
> | SMM*v1* | 50.49 ± 1.72 | 45.45 ± 1.91 | 40.89 ± 1.46 | 29.20 ± 1.79 | 6.07 ± 0.28 |
> | SMM*v2* | **51.34 ± 2.39** | **46.21 ± 2.09** | **41.61 ± 2.31** | **27.70 ± 2.20** | **5.93 ± 0.33** |
>
> **Avoid Early Overfitting**: We add a short warm-up phase where the model trains on the original masks before modulation begins.
>
> **Handling Overlaps**: Multi-class masks are separated channel-wise, so dilation in one class cannot affect another. This results in a multi-label setting where SMMv2 uses classwise BCE, while SMMv1 applies modulation only within the ESL term.
>
> **Computational Complexity**: Since our method introduces no new trainable parameters, the computational cost is fully determined by the base architecture. The overhead stems only from lightweight algebraic operations during modulation or loss computation, which is negligible relative to the forward/backward passes. In practice, this adds virtually no measurable complexity.
>
> We hope these points clarify the concerns raised. We will include all suggested updates in the updated version of the manuscript and remain open to further discussion.

---

### Meta-Review · Area_Chair_d4GQ · 2025-12-15

**Summary:**

The paper proposes a training-time strategy named SMM to address the imbalance between FP and FN in image segmentation tasks. The authors reported that FNR often exceeds FPR in domains like medical imaging or thin-structure segmentation. The method involves dilating FN regions and merging them into ground truth masks during training to force the model to capture missed areas.

However, the consensus of reviewers is that the technical novelty and empirical strength are insufficient, with concerns about limited improvements, limited architectural coverage (mostly U-Net/SegNet), and unclear positioning vs established loss designs.

**Reviewer Concerns:**

Addressed (or partially addressed) concerns:

1. more comprehensive baselines: The authors added comparisons against Tversky Loss and Focal Loss in the rebuttal phase

2. Computational cost: The authors clarified that the method introduces negligible overhead as it does not add trainable parameters.

unresolved concerns:

1. Contributions are incremental: all reviewers view the core mechanism as closely related to prior label modulation / cost-sensitive training, without a compelling theoretical framing or a clearly delineated unique benefit.

2. Evidence remains limited and overclaimed: Even with added analyses, the improvements are often modest and the “consistently outperforming SOTA” narrative is not convincingly supported across datasets/metrics; significance is mixed, and some reported metrics move in opposing directions.

3. Scope limited and generalization is not validated: Concerns remain about evaluating only small 2D datasets while motivating medical imaging broadly (often volumetric). The rebuttal states 3D extension is “natural” but does not provide validation.

**Reviewer Scores:**

The ratings of reviewers are unlikely to change, since their main concerns are only partially addressed

---

### Decision · Program_Chairs · 2026-01-26

Reject